# Chitosan Nanoparticles Containing Lipoic Acid with Antioxidant Properties as a Potential Nutritional Supplement

**DOI:** 10.3390/ani12040417

**Published:** 2022-02-10

**Authors:** Katrin Quester, Sarahí Rodríguez-González, Laura González-Dávalos, Carlos Lozano-Flores, Adriana González-Gallardo, Santino J. Zapiain-Merino, Armando Shimada, Ofelia Mora, Rafael Vazquez-Duhalt

**Affiliations:** 1Centro de Nanociencias y Nanotecnología, Universidad Nacional Autónoma de México, Km 107 Carretera Tijuana-Ensenada, Ensenada 22860, Mexico; quester@cnyn.unam.mx (K.Q.); santino.zapiain@gmail.com (S.J.Z.-M.); 2Laboratorio de Rumiología y Metabolismo Nutricional (RuMeN), FES-C, Universidad Nacional Autónoma de México, Blvd. Juriquilla 3001, Querétaro 76230, Mexico; sarahi.r.g.200@gmail.com (S.R.-G.); lauragdavalos@yahoo.com.mx (L.G.-D.); lozancarlos2020@gmail.com (C.L.-F.); shimada@unam.mx (A.S.); ofemora66@unam.mx (O.M.); 3Instituto de Neurobiología, Universidad Nacional Autónoma de México, Blvd. Juriquilla 3001, Querétaro 76230, Mexico; gallardog@unam.mx

**Keywords:** antioxidant, cell internalization, chitosan nanoparticles, everted intestine, lipoic acid

## Abstract

**Simple Summary:**

Alfa-lipoic acid (ALA) is an important antioxidant that could be added to animal feed as a nutritional supplement. To improve its stability in the digestive system, ALA was encapsulated in chitosan nanoparticles. The nanoparticles containing ALA were stable in stomach-like conditions and were able to cross the intestinal barrier. Chitosan-based nanoparticles seem to be an attractive administration method for antioxidants, or other sensible additives, in food.

**Abstract:**

The addition of the antioxidant α-lipoic acid (ALA) to a balanced diet might be crucial for the prevention of comorbidities such as cardiovascular diseases, diabetes, and obesity. Due to its low half-life and instability under stomach-like conditions, α-lipoic acid was encapsulated into chitosan nanoparticles (Ch-NPs). The resulting chitosan nanoparticles containing 20% *w*/*w* ALA (Ch-ALA-NPs) with an average diameter of 44 nm demonstrated antioxidant activity and stability under stomach-like conditions for up to 3 h. Furthermore, fluorescent Ch-ALA-NPs were effectively internalized into 3T3-L1 fibroblasts and were able to cross the intestinal barrier, as evidenced by everted intestine in vitro experiments. Thus, chitosan-based nanoparticles seem to be an attractive administration method for antioxidants, or other sensible additives, in food.

## 1. Introduction

Reactive oxygen species (ROS) are highly reactive molecules generated as by-products of oxygen metabolism in aerobic organisms [1]. Several diseases, such as hypertension and diabetes, amongst others, are associated with ROS generation [2,3]. Aging is one of the known interrupters of the balance between ROS generation and antioxidant defenses [4], and an age-related increase in oxidative damage to DNA, proteins and lipids was shown [5], particularly in cardiac and vascular tissues [6]. To slow down this oxidation-related damage, the antioxidant vitamins A, C, and E are popular nutritional supplements. However, clinical studies have failed to confirm their benefits in the prevention of cardiovascular diseases [7,8,9].

α-Lipoic acid (ALA), also known as thioctic acid or 1,2-dithiolane-3-pentanoic acid, is a natural antioxidant found in plants as well as in animal cells, where it is commonly found in mitochondria and acting as co-factor for a number of enzyme complexes that are involved in energy generation. Both existing forms of ALA—the oxidized (disulfide) and the reduced (dithiol; dihydrolipoic acid or DHLA) forms—showed antioxidant properties, and the ALA/DHLA couple is often referred to as the “universal antioxidant” due to its ability to not only improve but also restore intrinsic antioxidant systems after quenching free radicals [10,11,12,13]. Besides its great antioxidant potential, ALA was shown to efficiently remove heavy metals from the blood stream [11,13,14]. ALA has gained considerable attention as a dietary supplement because of its antioxidant activity and considerable anti-aging, anti-inflammatory, detoxifying, cognitive, cardiovascular, anti-cancer, and neuroprotective properties [15]. Some studies indicate that ALA might play an important role in the treatment of severe diseases such as diabetes mellitus [16,17,18], Alzheimer’s disease [19,20,21,22,23,24,25], obesity [26,27,28,29,30], multiple sclerosis [31,32,33], and schizophrenia [34,35,36].

Our research group previously demonstrated that broilers fed a diet supplemented with 40 mg/kg ALA for 7 weeks had significantly decreased liver levels of thiobarbituric acid reactive substances and hydroxyl radicals, whereas their total glutathione pools were increased compared to birds fed the same diet but without ALA [37]. We also demonstrated that the mRNA expression levels of the genes encoding four enzymes involved in energy metabolism (glutathione S-transferase theta 1 (GSTT1), oxoglutarate (alpha-ketoglutarate) dehydrogenase (OGDH), pyruvate dehydrogenase (lipoamide) alpha 1 (PDHA1), and dihydrolipoamide S-succinyl transferase (DLST)), as well as those encoding sirtuins 1 and 3, showed significant decreases in ALA-treated broilers compared to untreated controls (*p* < 0.01). These results confirm that the responses of broilers to ALA were associated with the down-regulation of certain liver enzymes, especially those involved in glucose metabolism and the tricarboxylic acid (TCA) cycle [38].

Nevertheless, the amount of this antioxidant synthesized by the organism does not meet bodily needs [39] unless supplementarily administered through diet. In addition, ALA presents a low half-life and bioavailability due to hepatic degradation, reduced solubility, and instability in the stomach environment [40]. Therefore, improving ALA administration is the main challenge, and the encapsulation of ALA into biocompatible nanoparticles seems to be a promising alternative.

Chitosan is a polymer produced after deacetylation of the natural polymer chitin, which is the main component of the shells of crabs and other crustaceans. The United States Food and Drug Administration (FDA) and the European Commission, among others, have approved chitosan as safe for use in food and drugs. Chitosan has been widely used for the encapsulation of proteins and therapeutical enzymes [41,42,43,44] or as biocatalytic nanoparticles [45]. Importantly, among the biopolymers used as nanocarriers, it has attracted significant attention since it is positively charged [46]. Therefore, this polysaccharide can be customized for a wide variety of chemical and ionic side-chain reactions [44]. Chitosan nanoparticles are known to easily achieve a sustained slow release of the cargo as well as increase bioavailability and therapeutic efficiency, while being biodegradable and non-toxic [47]. The production process for nanoparticles such as chitosan NPs could be improved to gain the advantages of ease of production, high yield at low cost, and larger loading capacity [48,49,50,51,52].

In this study, chitosan-α-lipoic acid nanoparticles (Ch-ALA-NPs) with antioxidant activity were synthesized, and their stability under stomach-like conditions was assayed. Furthermore, the cell internalization of chitosan nanoparticles in intestinal cells and fibroblasts and the ability to cross the intestinal mucosal barrier in vitro by using the everted intestine technique were demonstrated.

## 2. Materials and Methods

### 2.1. Synthesis of Chitosan-α-Lipoic Acid Nanoparticles (Ch-ALA-NPs)

The chitosan nanoparticles were prepared based on the protocol described by [53] with the following modifications: Chitosan obtained from the deacetylation of shrimp shell chitin was purchased from Future Foods (Tlalnepantla de Baz, Mexico). The preparation showed 90.4% deacetylation, a viscosity of 54 mPa·s in 1% acetic acid, and an average molecular weight of 290 kDa. The chitosan solution was prepared by dissolving 2.5 mg mL^−1^ of chitosan in 2% acetic acid while stirring at approximately 200–300 rpm overnight at room temperature, and then centrifuging for 15 min at 8000× *g* at 4 °C. The resulting supernatant was again centrifuged for 20 min at 11,000× *g* at 4 °C. Then, the supernatant was filtered through a 1.2 μm membrane, and the pH was adjusted to 4.5.

Nanoparticle synthesis was achieved using an automatic pipet system designed by our research group and equipped with an insulin syringe to control the addition of reagents at a constant rate. The reagent addition was performed at room temperature and constant agitation of approximately 800 rpm. To 5 mL of the chitosan solution, 200 μL of α-lipoic acid solution (6 mg mL^−1^ in 20% ethanol) were added at 0.09 mm/s. After 15 min, 1 mL of tripolyphosphate pentasodium (TPP; Sigma-Aldrich, St. Louis, MO, USA) solution (0.25 mg mL^−1^) was added at 0.09 mm/s and left to react for 1 h. Then, 100 μL of 2.5% glutaraldehyde (Sigma-Aldrich, USA) were added at 0.09 mm/s and again left for 1 h. Afterwards, the solution was centrifuged for 30 min at 2000× *g* at room temperature (Sorvall Legend RT centrifuge) to remove chitosan aggregates. The resulting supernatant was then ultracentrifuged for 2 h at 60,000× *g* at 4 °C (Beckman Coulter Optima XPN-100 ultracentrifuge). Subsequently, the pellet, containing the chitosan-α-lipoic acid nanoparticles (Ch-ALA-NPs), was resuspended in 100 mM phosphate butter (pH 6), filtered through a 0.2 μm membrane, and stored at 4 °C until further modification.

### 2.2. Synthesis of Chitosan-Green Fluorescent-Nanoparticles Ch-GFP-NPs

Chitosan nanoparticles labeled by encapsulating carboxylated green fluorescent protein (GFP) were synthetized as previously reported [44]. In brief, GFP was reacted with malonic acid in the presence of carbodiimide (EDC) (N-(3-dimethyl-aminopropyl)-N′-ethylcarbodiimide hydrochloride) and N-hydroxysuccinimide (NHS). The amount of malonic acid was twice the amount of free carboxylic groups on the protein surface on a molar basis, and the amounts of EDC and NHS were ten and five times the amount of malonic acid, respectively. The mixtures were allowed to react for 3 h in rotation at room temperature and then dialyzed against MES (50 mM, pH 6). The encapsulation of carboxylated GFP was performed by ionic gelation with TPP as described above.

### 2.3. Morphological Analyses of the Nanoparticles

The hydrodynamic diameter and Zeta potential of all preparations were determined by the DLS technique on a Zetasizer NanoZS (Malvern, UK). To confirm the size and shape of the nanoparticles, high-resolution transmission electron microscopy (HR-TEM) was performed using a JEOL JEM-2010. Five μL of a 1000 times diluted sample of the nanoparticles were placed onto a carbon-coated 300 mesh copper grid (Ted Pella, Inc.; Redding, CA, USA) and then stained with phosphotungstic acid (PTA).

### 2.4. Cargo Capacity and Antioxidant Activity of the Ch-ALA-NPs

To estimate the amount of encapsulated α-lipoic acid, HPLC analyses were performed using the supernatant after ultracentrifugation. The resulting concentration of free α-lipoic acid was subtracted from the initial α-lipoic acid concentration used for nanoparticle synthesis.

Antioxidant activity of free and Ch-ALA-NPs was determined by using the inhibition of pyrogallol autooxidation method reported by [54], with some modifications. First, 50 mM Tris-HCl buffer (pH 8.2, containing 1 mM EDTA) was oxygenized with air. Pyrogallol (2 mM) was dissolved in this oxygenized buffer. The capacity of the free ALA and Ch-ALA-NPs to inhibit the spontaneous pyrogallol oxidation in the presence of atmospheric oxygen was determined using equivalent concentrations of the nanoparticles and free α-lipoic acid and monitored in a UV-vis spectrophotometer (Lambda 25, Perkin Elmer, Waltham, MA, USA) at 420 nm.

### 2.5. Release of α-Lipoic Acid from the Ch-ALA-NPs

The release of α-lipoic acid from the chitosan nanoparticles was determined in HCl (pH 2) at 37 °C for 3 h, to simulate the stomach environment. A suspension of Ch-ALA-NPs (10 mL) was added to a dialysis membrane with a molecular weight cut-off (MWCO) of 14 kDa. The dialysis membrane containing the nanoparticle suspension was transferred into 20 mL of HCl (pH 2). Under constant agitation at room temperature, 1 mL samples of HCl were measured at 330 nm at different time points to determine the corresponding amounts of released α-lipoic acid. The released ALA was quantified by using a standard curve.

### 2.6. Fluorophore Labeled Nanoparticles

In order to monitor and quantify the nanoparticles containing α-lipoic acid in the everted intestine experiments, the nanoparticles were conjugated with a fluorophore (Fluorescein isothiocyanate, FITC), and produced as follows: One mL of ALA solution in isopropanol (100 mg/mL) was added to 35 mL solution of chitosan (2%) in acetic acid, pH 5.5. The ALA solution was added at a rate of 0.09 mm/s using the automatic pipet system designed by our research group and described above. After 15 min, 250 μL of a solution of FITC in methanol (10 mg/mL) were added at a rate of 0.03 mm/s. Then, 7 mL of 0.25 % TPP were added at a rate of 0.09 mm/s, and the mixture was shaken for 1 h to produce Ch-ALA-FITC-NPs by ionic gelation. Finally, 1 mL of glutaraldehyde (2.5%) was added with the same system and shaken for 1 h. The nanoparticle suspension was centrifuged at 3500 rpm for 30 min to eliminate the bigger particles, and the supernatant was then centrifuged at 60,000× *g* for 2 h. The pellet containing the Ch-ALA-FITC-NPs was resuspended in Milli Q water.

### 2.7. Internalization of Ch-GFP-NPs into Fibroblasts 3T3-L1

Fibroblast 3T3-L1 (CL-173, ATCC) cells (1 × 10^5^) were cultivated in DMEM containing 10% fetal bovine serum, 1% penicillium and 1% streptomycin in 6-well plates (Corning, NY, USA) containing microscope glass slides and incubated at 37 °C and 5% CO_2_. Once they reached a confluence of 80–85%, Ch-GFP-NPs were added in different concentrations (5, 10, 20 and 50 μL mL^−1^) to the medium and incubated for 24 and 48 h. Afterward, the glass slides, on which cells adhered, were washed with PBS, stained for 15 min with MitoTracker Red CMXRos (Invitrogen, Waltham, MA, USA), and fixed with 4% PFA. Cell nuclei were stained with 4′,6-diamidino-2-phenylindole (DAPI) (Invitrogen, Carlsbad, CA, USA). Cells were then analyzed with a confocal laser scanning microscope (LSM 780, Carl Zeiss Inc., Oberkochen, Germany) using a DPSS laser with emission at 561 nm and a HeNe laser with emission at 633 nm. Obtained images were analyzed using the software ZEN Blue Edition (Zeiss, Jena, Germany) and ImageJ (NIH, Bethesda, MD, USA).

The 3D reconstruction was performed with Amira software (Thermo Fisher Scientific, Waltham, MA, USA) by visualizing, processing, and analyzing data obtained by 780 LSM confocal microscope (Zeiss, Jena, Germany). Nuclei were seen in blue and mitochondria in red, while nanoparticles were in green; the observed signal was processed by the isosurface and volume rendering module. This visualization was carried out using immersive reality within the Cave Automatic Virtual Environment (CAVE) of the LAVIS-UNAM laboratory.

### 2.8. Intestine Barrier Crossing of Ch-ALA-FITC-NPs by In Vitro Everted Intestine

Intestinal tissues were taken from Wistar rats and chickens (*Gallus gallus*) (donated from a local farm), to compare NP translocation through the intestine barrier in two species by everted intestine experiments. Wistar rats were housed and used at Universidad Nacional Autónoma de México (UNAM) according to regulations of the Mexican government regarding the use of laboratory animals for research purposes (CICUAL-UNAM, NOM-062-ZOO-1999 4.2.2). Animals were sacrificed with CO_2_ and by cervical dislocation by trained personnel. This protocol was approved by the Institute’s Research Ethics Committee (Comité de Ética en la Investigación, INB-UNAM) with register #065. Intestinal tissue was obtained from 20–21 day old male rats, and 6-month-old chickens. Twelve cm of intestine from each rat or 10 cm from each chicken were dissected and placed in a Petri dish containing Krebs-Ringer buffer at 37 °C. Immediately, the eversion of the intestine was carefully realized using a Pasteur pipette. The intestinal content was removed, and the tissue was washed with the same buffer. The lower end of the intestine was closed using a nylon thread, 600 μL Krebs-Ringer buffer were added, and the other end of the intestine was closed as well. The sealed intestine was then transferred into a 15 mL Falcon tube, previously gasified with CO_2_, containing 50 μL/mL Ch-ALA-FITC-NPs in Krebs-Ringer buffer, and incubated for 60 min at 37 °C under agitation. Subsequently, the intestinal content was centrifuged for 1 min, and fluorescence of the supernatant was spectrofluorimetrically measured at an excitation of 495 nm and an emission of 519 nm. Control experiments were carried out in the same conditions but without adding nanoparticles. The percentage of nanoparticle internalization was estimated by comparing the fluorescence of the samples to the fluorescence of the initial nanoparticle solution. All experiments were performed in triplicate.

### 2.9. Statistical Analysis

Significant differences in all determinations were analyzed by one-way ANOVA test, followed by post hoc Tukey rank test. Statistical significance was set at *p* < 0.05 and expressed as compact letter display. Statistical analysis was performed on five independent replicates using the software STATISTICA 8.0™ (StatSoft Inc., Tulsa, OK, USA).

## 3. Results

Chitosan nanoparticles were produced by ionic gelation in the presence of ALA. Preliminary experiments to determine the maximal ALA loading in the chitosan nanoparticles were performed. A fixed amount of chitosan (12.5 mg) and different concentrations of ALA from 0.5 to 5 mg were used to produce chitosan ALA-loaded nanoparticles (Ch-ALA-NPs). The ALA content was then determined spectrophotometrically. Higher ALA concentrations than 5 mg induced precipitation and no nanoparticle formation.

The aim of this work was to produce chitosan nanoparticles as carriers for ALA. Four different preparations were synthetized: control nanoparticles without ALA (Ch-NPs), chitosan nanoparticles containing green fluorescent protein (Ch-GFP-NPs) to monitor cell internalization of nanoparticles by confocal microscopy, FITC-modified and -containing ALA particles (Ch-ALA-FITC-NPs) to evaluate intestinal barrier crossing by the everted intestine technique by fluorescence, and the ALA-containing chitosan nanoparticles (Ch-ALA-NPs) (Table 1). The nanoparticles formed with a Ch:ALA ratio of 2.5:1 showed an ALA content of 20% *w*/*w*, and this ratio was selected for further experiments.

Dynamic light scattering (DLS) analyses (Table 1) revealed a hydrodynamic diameter of approximately 44.1 (±20.8) nm, which was confirmed by high-resolution transmission electron microscopy (HR-TEM) (Figure 1). The nanoparticle size was half that of the nanoparticles without ALA (Ch-NPs) and could be due to a strong interaction between the carboxylic group of ALA and the amino group of glucosamine monomers of chitosan. This interaction was also supported by the reduction of zeta potential from 49 mV to 32 mV when the ALA was loaded into nanoparticles (Table 1).

The antioxidant activity of both free and nano-encapsulated ALA was also determined (Figure 2). The inhibition of the spontaneous oxidation of pyrogallol in the presence of atmospheric oxygen was used as the antioxidation assay. The antioxidant activity was not significantly affected by the encapsulation, Ch-ALA-NPs being slightly more efficient than free ALA, and both preparations were able to completely inhibit this reaction at the same ALA equivalents. Chitosan-only nanoparticles (Ch-NPs) showed some antioxidant activity in the oxidation of pyrogallol.

As mentioned above, the low half-life and bioavailability of ALA, as well as its instability in the stomach, represent disadvantages for its administration as a dietary supplement. Thus, the stability of Ch-ALA-NPs and liberation of ALA in a stomach-like environment (HCl solution at pH 2) were determined. Free ALA was measured via UV-vis spectrophotometry every 5 min for 3 h (Figure 3). A sustained release of ALA was found, but the percentage of released ALA from the NPs was very low according to the total ALA content in the NPs, implying that the Ch-ALA-NPs remained stable and intact in the stomach until further intestinal absorption. Unmodified chitosan is fully soluble at low pH. Nevertheless, the glutaraldehyde crosslinking of chitosan nanoparticles made them stable even in very acidic conditions, thus retaining the ALA inside.

Cell internalization of nanoparticles in 3T3 cells of mouse embryonic fibroblasts was assayed using synthesized fluorescent NPs containing the green fluorescent protein (Ch-GFP-NPs) (Figure 4). These fluorescent chitosan nanoparticles showed hydrodynamic diameter and zeta potential similar to non-labeled nanoparticles (Ch-NPs). A culture of 3T3 cells of mouse embryonic fibroblasts was incubated with different concentrations of Ch-GFP-NPs; after 48 h of incubation, the cells were washed three times with PBS and analyzed by confocal laser scanning microscopy (see Appendix A). The cell nuclei were stained with DAPI, and the mitochondria with MitoTracker Red. The confocal analysis showed clear evidence of chitosan nanoparticle cell internalization.

The capacity of chitosan nanoparticles loaded with α-lipoic acid to cross the intestinal barrier was evaluated by using the everted intestine technique. To achieve this, the nanoparticles were labeled with FITC (Ch-ALA-FITC-NPs) to be monitored and quantified. Because the FITC was covalently bonded to the amino groups of glucosamine monomers of chitosan, the zeta potential was reduced to 28 mV (Table 1), but still positive. The everted intestine technique is widely used to evaluate intestinal absorption of nutrients, drugs, and toxicants [55,56,57]. Everted intestines from rats and chickens were incubated with Ch-ALA-FITC-NPs, and the fluorescence was measured in samples taken from both inside and outside of the everted intestine (Figure 5). Although the absorption rate was low (~0.5%/h), a significant level of fluorescence could be observed compared to the control without NP treatment. 

## 4. Discussion

α-Lipoic acid is synthesized in human and animal cells at a low rate, and thus it should be consumed as a dietary supplement. However, ALAs’ instability in the stomach and low bioavailability represent challenges for efficient administration to the animal organism [40]. To overcome these obstacles, ALA was encapsulated in chitosan nanoparticles. This natural biopolymer is approved by most agencies around the world as safe for use in food and drugs, and it is known to easily achieve a sustained slow release of its cargo as well as increase bioavailability and therapeutic efficiency [47].

Chitosan-based nanoparticles containing ALA were produced (Figure 1 and Table 1). Due to the solubility of chitosan at low pH, the nanoparticles were stabilized by glutaraldehyde crosslinking. However, the glutaraldehyde molecules were covalently conjugated to the glucosamine moieties and not in free form, and thereby reducing their potential toxicity. The antioxidant activity of ALA was slightly higher when encapsulated in chitosan nanoparticles, while the NPs without ALA showed some antioxidant activity (Figure 2). The slight antioxidant capacity of chitosan is well known [58,59,60].

The chitosan nanoparticles were internalized by mouse embryonic fibroblasts (Figure 4 and Appendix A). Chitosan nanoparticle uptake by different cells has been clearly demonstrated [61,62,63]. The Ch-ALA-NPs showed a positive Zeta potential of 35.7 ± 3.1 at pH of 6. The positive charge promoted the cell internalization rate in different cell lines, increasing the cellular uptake of NPs. In addition, an intracellular trafficking study indicated that positively charged NPs exhibit perinuclear localization and escape from lysosome [60].

Using the everted intestine technique, the capacity of chitosan nanoparticles containing α-lipoic acid to cross the intestinal barrier was demonstrated (Figure 5). Nanoparticles in a size range of 0.1 to 100 nm can quickly enter the digestive tract [64]. Due to their size (84 nm), the chitosan nanoparticles showed the ability to cross the mucosal barrier and interact with the subjacent absorbent epithelial tissue [65,66]. On the other hand, chitosan is a polymer that can help the absorption and increase the permeation capacity of drugs through the paracellular pathway by the reversible opening of the narrow epithelial bonds [67]. In addition, chitosan is muco-adhesive, and thus can adhere to the intestinal mucosa through ionic interactions between its positively charged amino groups and the negative charges found on the surfaces of intestinal cells, promoting intestinal absorption [68]. It is well known that chitosan nanoparticles can increase the drug concentration at the absorption sites [65,66]. In addition, it has been demonstrated that chitosan nanoparticles increase the intestinal absorption in vitro [69] of diverse bioactive compounds including, phenolic compounds [70], catechins and epigallocatechin from green tea [71], resveratrol and cumarin-6 [72].

The results obtained in the present study indicate that the administration of α-lipoic acid in the form of Ch-ALA-NPs as a supplement to a balanced diet might be beneficial for, the prevention and/or treatment of severe diseases such as cardiovascular diseases, diabetes and obesity, amongst others. ALA may play an important role in the treatment and/or prevention of diverse severe diseases including diabetes, Alzheimer’s, multiple sclerosis, schizophrenia, and obesity [16,17,21,22,25,26,27,28,31,32,35].

## 5. Conclusions

Chitosan nanoparticles containing α-lipoic acid with antioxidant activity were efficiently synthesized in a size average of 44 nm. The nanoparticles proved to be stable under stomach-like conditions for up to 3 h. Furthermore, the cell internalization of chitosan nanoparticles into fibroblasts cells was demonstrated. Finally, the nanoparticles were able to cross the intestinal barrier as confirmed by the everted intestine technique, suggesting an efficient method to supply ALA. From our results, we concluded that chitosan-based nanoparticles containing ALA were stable in stomach-like conditions and able to cross the intestinal barrier and release their antioxidant cargo. Thus, the use of α-lipoic acid in a nanoparticulate form as a dietary supplement seems to be an attractive method of administration. However, the results shown here are from in vitro experiments; thus, to determine potentially better bioavailability than the native compound, adequately designed pharmacokinetic trials in animal models should be carried out.

## Figures and Tables

**Figure 1 animals-12-00417-f001:**
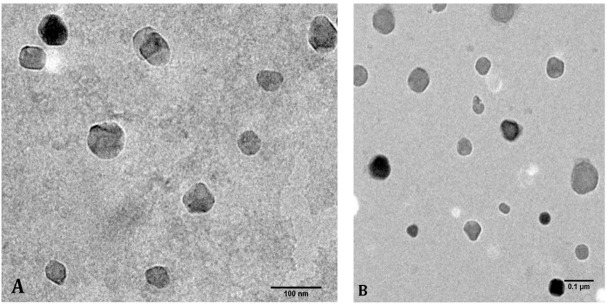
HR-TEM images of synthesized Ch-ALA-NPs reveal an average diameter of 44 nm. Scale bars are 100 nm in (**A**), and 0.1 μm in (**B**).

**Figure 2 animals-12-00417-f002:**
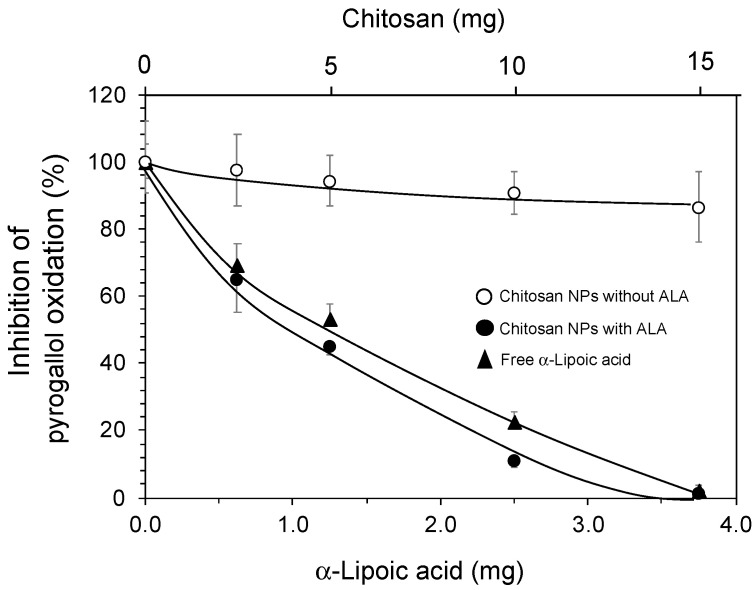
Antioxidant activity of Ch-NPs, CH-ALA-NPs and free ALA. The antioxidant activity was determined by using the inhibition of pyrogallol autooxidation method reported by Marklund and Marklund [54]. The average values from five independent experiments and the standard deviation as error bars are shown.

**Figure 3 animals-12-00417-f003:**
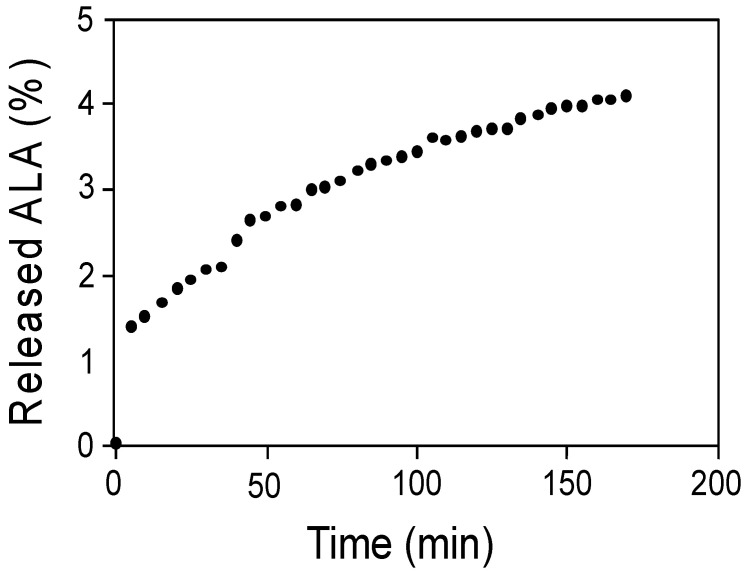
ALA release from Ch-ALA-NPs in a stomach-like environment (HCl, pH 2).

**Figure 4 animals-12-00417-f004:**
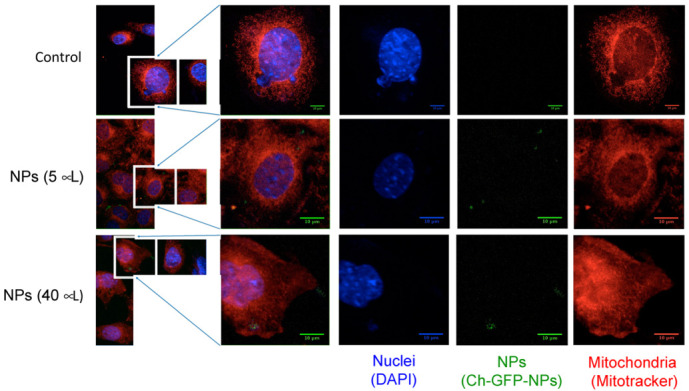
Cell internalization of chitosan nanoparticles containing GFP into 3T3 cells of mouse embryonic fibroblasts. Ch-GFP-NPs were incubated for 48 h and then washed three times. The cell nuclei were stained with DAPI, and the mitochondria with MitoTracker Red. The nanoparticles were detected by emission at 561 nm.

**Figure 5 animals-12-00417-f005:**
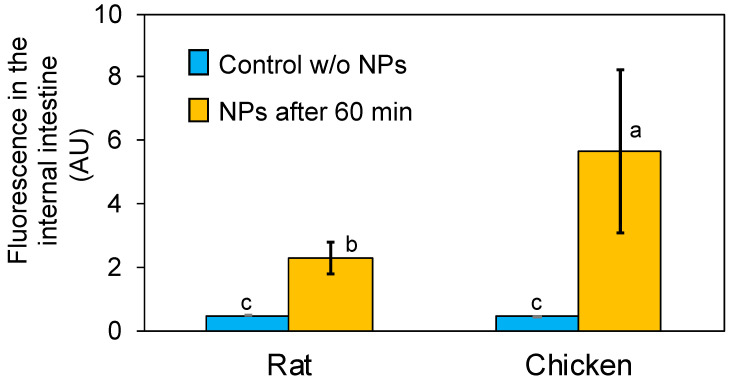
Internalization through the intestinal barrier of Ch-ALA-FITC-NPs assayed in vitro by the everted intestine technique. Fluorescence in arbitrary units (AU) in the control experiments without nanoparticles and after 60 min incubation with Ch-ALA-FITC-NPs. The experiments were carried out at 37 °C, and average values from five independent experiments and the standard deviation as error bars are shown. Significant differences in all determinations were analyzed by one-way ANOVA test, followed by post hoc Tukey rank test. Statistical significance was set at *p* < 0.05. The meaning of a,b,c in figure is the statistical significance expressed as compact letter display.

**Table 1 animals-12-00417-t001:** Hydrodynamic diameter and Zeta potential of different preparations of chitosan-based nanoparticles.

Nanoparticle Preparation	Hydrodynamic Diameter (nm)	Zeta Potential (mV)
Ch-NPs	88.4 (±28.2) ^a^	49 (±1.9) ^a^
Ch-GFP-NPs	96.7 (±35.2) ^a^	45 (±2.2) ^b^
Ch-ALA-NPs	44.1 (±20.8) ^b^	32 (±0.8) ^c^
Ch-ALA-FITC-NPs	84.6 (±28.2) ^a^	28 (±0.2) ^d^

^a,b,c,d^ Significant differences from four independent determinations were analyzed by one-way ANOVA test, followed by post hoc Tukey rank test. Statistical significance was set at *p* < 0.05.

## Data Availability

Not applicable.

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
