# Peer review of "Chitosan Nanoparticles Containing Lipoic Acid with Antioxidant Properties as a Potential Nutritional Supplement"

_animals, 2022, doi:10.3390/ani12040417_

Round 1

Reviewer 1 Report

In this interesting study the authors show that chitosan nanoparticles containing ALA, a natural antioxidant, were stable in stomachal conditions, were able to cross the intestinal barrier and to release their antioxidant cargo. Even if these are in vitro experimental data, the use of ALA in this kind of nanoparticulate form seems to be attractive and deserves to be further investigated with pharmacokinetic studies in preclinical models. The study is described in a very fluent and captivating way.

General considerations:

1) To better mimic the stomachal conditions pepsin had to be added to HCl (pH 2).

2) For future in vivo study perhaps it would be better to develop a nanoparticles synthesis method independent of glutaraldheyde.

Minor revisions:

1) In M&M section, 2.8 paragraph, "..in vitro everted intestine" the authors have to specify the control conditions utilized and reported in Figure 5.

3) Figure 1. Scale bars are not visible in the images so you have to report them in the caption.

4) Affiliation for Carlos Lozano-Flores is missing. 

5) Lane 236_ replace the comma with a period after (Figure 1)

6) Lane 224_"four" and not "for"

7) Lane 341_ "their" and not "its"

Author Response

We would like to thank all reviewers and the editor for the comments and suggestions that doubtless, have improved our work. The manuscript has been modified as follows:

Editor comments

According to the journal style the Results and Discussion sections have been split up.

Reviewer 1

In this interesting study the authors show that chitosan nanoparticles containing ALA, a natural antioxidant, were stable in stomachal conditions, were able to cross the intestinal barrier and to release their antioxidant cargo. Even if these are in vitro experimental data, the use of ALA in this kind of nanoparticulate form seems to be attractive and deserves to be further investigated with pharmacokinetic studies in preclinical models. The study is described in a very fluent and captivating way.

  1. We are glad about the reviewer opinion on our work, and we are thankful for his/her suggestions.

General considerations:

1) To better mimic the stomachal conditions pepsin had to be added to HCl (pH 2).

  1. We have starting in vivo experiments to evaluate the real stability of nanoparticles. These results will be part of a future manuscript.

2) For future in vivo study perhaps it would be better to develop a nanoparticles synthesis method independent of glutaraldehyde.

  1. As mentioned in the text, the glutaraldehyde is needed to stabilize the chitosan nanoparticles, especially at low pH. However, the glutaraldehyde molecules are covalently conjugated and not in free form. Thus, the potential toxicity is largely reduced. A new paragraph has been included mentioning this.

Minor revisions:

1) In M&M section, 2.8 paragraph, "..in vitro everted intestine" the authors have to specify the control conditions utilized and reported in Figure 5.

  1. Control conditions have been included.

3) Figure 1. Scale bars are not visible in the images so you have to report them in the caption.

  1. The values of scale bars have been included in the figure legend.

4) Affiliation for Carlos Lozano-Flores is missing.

  1. The error has been corrected.

5) Lane 236_ replace the comma with a period after (Figure 1)

  1. The comma has been replaced.

6) Lane 224_"four" and not "for"

  1. The error has been corrected.

7) Lane 341_ "their" and not "its"

  1. The error has been corrected.

Reviewer 2 Report

This manuscript will be eligible to be considered for publication in Animals with minor revision.

Line 163-180: When proprietary brands are used in research, include the brand name and the name, city and country of origin of the manufacturer in parentheses after the first mention of the generic name in the Methods section.

There is a need to improve the resolution of the figures (Fig. 1 and 3), and to accurately represent statistically significant differences of results in tables and figures and to write details in figure legends.

Author Response

We would like to thank all reviewers and the editor for the comments and suggestions that doubtless, have improved our work. The manuscript has been modified as follows:

Editor comments

According to the journal style the Results and Discussion sections have been split up.

Reviewer 2

This manuscript will be eligible to be considered for publication in Animals with minor revision.

Line 163-180: When proprietary brands are used in research, include the brand name and the name, city and country of origin of the manufacturer in parentheses after the first mention of the generic name in the Methods section.

  1. The location of the brands in the methods section have been included.

There is a need to improve the resolution of the figures (Fig. 1 and 3), and to accurately represent statistically significant differences of results in tables and figures and to write details in figure legends.

  1. The resolution of figures has been improved, and the statistically differences were accurately represented.

Reviewer 3 Report

Manuscript animals-1573384 present the preparation and the characterization of an antioxidant, lipoic acid, loaded in the chitosan nanoparticles. The manuscript needs improvements before publication.

The manuscript is intended to be just a proof-of-concept because the described preparation process for chitosan nanoparticles loaded with lipoic acid is not feasible to scale up. Authors mentioned in L79-L81 that “nanoparticles like chitosan NPs offer the advantage of ease of production, high yield at low cost, and larger loading capacity.” This affirmation is not valid for their preparation process, which involves ultracentrifugation. My suggestion to the authors is to underline this in the Introduction and Results and Discussion sections.

L70-L71. Authors mentioned that “Chitosan, a natural deacetylated biopolymer product from chitin, the main component of the shell of crabs and other crustaceans” This proposition lacks the verb (is) and needs to be corrected to avoid readers misleading. Chitosan is not naturally deacetylated – the deacetylation is man-made. The chitin is also the main component of the shell of insects and other arthropods and is an important component of fungal cell walls.

Important information regarding the used chitosan is missing, i.e., origin/source, molecular mass, deacetylation degree. Also, for nanoencapsulation purposes, a deacetylation pattern is important. The authors must present  (a least) the origin/source, molecular mass, deacetylation degree. Without this essential information, the reproducibility of the experiments from manuscript animals-1573384 is difficult.

Figure caption provides redundant information, exceeding the “stand-alone” request, i.e., understanding a figure without reading the text. For example, citation of the used method is not usual in a figure caption – my suggestion is to remove it.

In the Author Contributions Section, the CRediT (Contributor Roles Taxonomy) must be (also) used.

Author Response

We would like to thank all reviewers and the editor for the comments and suggestions that doubtless, have improved our work. The manuscript has been modified as follows:

Editor comments

According to the journal style the Results and Discussion sections have been split up.

Reviewer 3

The manuscript is intended to be just a proof-of-concept because the described preparation process for chitosan nanoparticles loaded with lipoic acid is not feasible to scale up. Authors mentioned in L79-L81 that “nanoparticles like chitosan NPs offer the advantage of ease of production, high yield at low cost, and larger loading capacity.” This affirmation is not valid for their preparation process, which involves ultracentrifugation. My suggestion to the authors is to underline this in the Introduction and Results and Discussion sections.

R. The sentences in the introduction section has been modified to be more accurate.

L70-L71. Authors mentioned that “Chitosan, a natural deacetylated biopolymer product from chitin, the main component of the shell of crabs and other crustaceans” This proposition lacks the verb (is) and needs to be corrected to avoid readers misleading. Chitosan is not naturally deacetylated – the deacetylation is man-made. The chitin is also the main component of the shell of insects and other arthropods and is an important component of fungal cell walls.

R. The paragraph has been modified accordingly the reviewer suggestion to avoid misleading.

Important information regarding the used chitosan is missing, i.e., origin/source, molecular mass, deacetylation degree. Also, for nanoencapsulation purposes, a deacetylation pattern is important. The authors must present (a least) the origin/source, molecular mass, deacetylation degree. Without this essential information, the reproducibility of the experiments from manuscript is difficult.

R. The source and properties of the chitosan have been included.

Figure caption provides redundant information, exceeding the “stand-alone” request, i.e., understanding a figure without reading the text. For example, citation of the used method is not usual in a figure caption – my suggestion is to remove it.

R. The redundant information in the figure legends were removed.

In the Author Contributions Section, the CRediT (Contributor Roles Taxonomy) must be (also) used.

R. The CRediT system has been used for the Author Contribution Section.
